# Low-Cost, Open-Source, Emoncms-Based SCADA System for a Large Grid-Connected PV System

**DOI:** 10.3390/s22186733

**Published:** 2022-09-06

**Authors:** Luqman Ahsan, Mirza Jabbar Aziz Baig, Mohmmad Tariq Iqbal

**Affiliations:** Department of Electrical and Computer Engineering, Memorial University of Newfoundland, 230 Elizabeth Avenue, St. Johns, NL A1C 5S7, Canada

**Keywords:** SCADA, Emoncms, Internet of Things, Raspberry Pi, Emonhub, solar energy

## Abstract

This article describes a low-cost Supervisory Control and Data Acquisition (SCADA) system for a PV plant with local data logging. Typically, SCADA systems that are available on the market are proprietary (commercial), which are expensive and individually configured for a particular site. The main objective of this paper is to design a low-cost and open-source monitoring solution (hardware and software) to meet the requirements. The hardware used for this SCADA consisted of Arduino, Raspberry Pi, sensors, serial communication cables, and an open-source web view platform. This open-source platform manipulates, logs, and visualizes PV and environmental data. Emoncms runs on the Debian operating system. Field instruments were connected to two remote terminal units (RTUs). A PV array provided data to the RTU1, while an inverter output provided data to the RTU2, and the Raspberry Pi received the collected data in JSON format. As these data arrived, Emoncms used Emonhub as its main module, which refines data and then displays it on Emoncms’s WebView. The Raspberry Pi also stores data locally. Data logging was tested for 6 h, but the final results showed that data logging can last much longer. From an hour to a year, the data trend can be viewed on a user-friendly dashboard.

## 1. Introduction

Monitoring, control, and data acquisition are all referred to as SCADA. All of these functions comprise both hardware and software components. In addition to collecting, monitoring, and processing real-time data, it also allows industrial organizations to control processes in real-time from either local or remote locations. In a SCADA system, a human-machine interface (HMI) is used to interact with sensors and devices, and log files are generated. Industrial organizations depend on SCADA systems to eliminate downtime and increase efficiency while processing data and making smarter decisions. As depicted in Figure 1, a programmable logic controller (PLC) or remote terminal unit (RTU) is among the essential components of SCADA systems. A PLC or RTU is a microcomputer that communicates with objects of various types (a factory machine, a human-machine interface, a sensor, or an end device are some examples) and routes the information to a computer running SCADA software. As a result of SCADA software, data are processed, distributed, and analyzed, enabling operators to make important decisions as a result [1,2].

Finally, the SCADA system is used in some process and utility applications to enhance the performance of a system. As a result of global warming, carbon emissions have to be reduced and renewable energies should be used instead. As a result of this dilemma, solar energy greatly contributes to supplying energy to the electricity industry at an affordable and carbon-free price. Therefore, after completing the sizing [2] and dynamic modeling of a system [3], the SCADA of a system was designed for the real-time monitoring of a system which provides user-friendly monitoring along with the local data logging that ultimately provides a secure and reliable system.

Throughout this article, we have organized the information in the following way. A detailed literature review is presented in Section 2 of the paper. There is a complete system description in Section 3 and a detailed component description in Section 4 of this article. Section 5 describes the implementation methodology, while Section 6 and Section 7 present the results and prototype design and discussions, respectively. This article is concluded in Section 8.

## 2. Literature Review

During this research, a comprehensive literature review was conducted, and some of the useful sources are outlined here. In [4], the vulnerability of SCADA systems to cyberattacks is discussed by the authors. The authors emphasize that SCADA systems’ cybersecurity is becoming increasingly important as industrial digitization increases. They explored methods for distinguishing genuine cyberattacks from equipment faults. In particular, the authors examined “replay attacks” (RA), which are a relatively complex cyberattacks. Mathematical formalisms, comprehensive analysis, and experiments on rotating machinery are the tools used in this study to differentiate between the equipment faults and the RA cyberattacks. In another study, the authors say that real-time information has become imperative for managing renewable energy assets with the rapid growth of renewable energy generation worldwide. The authors claim that as of now, the renewable energy sources have not been monitored in real time using cost-effective condition monitoring techniques to ascertain the optimal utilization of these invaluable. The authors presented an application of Supervisory Control and Data Acquisition (SCADA) using the Internet of Things to monitor a hybrid system combining wind, photovoltaics, and energy storage systems. The authors utilized ThingSpeak for the real-time monitoring of electrical parameters and the system was integrated with Matlab/Simulink using the KEPServerEX client. The authors considered the cost effectiveness of the system by implementing low-cost electronic components. The proposed system enabled system operators to access the system remotely. They claimed that through simulations and experiments that the proposed system was proven to be feasible, reliable, and cost-effective [5]. In [6], a SCADA system was used to remotely control and monitor inverters with grid connectivity. To maintain stable energy prices and network stability, utility providers require proper monitoring and control of grid-connected inverters. The number of grid-connected inverters increases with the number of batteries tied with the power system, and the authors considered these crucial aspects as a part of this research. Testing results led to significant improvements in the SCADA system, and its main component is an Internet of Things (IoT) server. In response to changes in energy prices, renewable energy generation fluctuates, and the proposed SCADA system in this study could automatically manage the inverter to obtain economic benefits. The authors also used an algorithm to enhance economic benefits.

The authors of [7] presented a low-cost and open-source SCADA system by using the Internet of Things (IoT) and advanced SCADA system design. The proposed system applied an ESP32 micro controller, voltage sensors, current sensors, a ThingsBoard IoT server, and HMIs. For data transfer, the authors configured Message Queuing Telemetry Transport (MQTT). The proposed system by the authors was successfully tested and was claimed to be feasible as a low-cost and open-source SCADA system for the remote monitoring and control of a small system. As a part of [8], the authors used an ESP32 and the Arduino IoT Cloud to implement an open source, low-cost, IoT-based SCADA system for a rural Base Transceiver Station (BTS). A Wi-Fi network was used as a communication channel to transmit data from a current, voltage, temperature, and humidity sensor to the Arduino IoT Cloud. To monitor and control the system, widget-based dashboards on the Arduino IoT Cloud were used in the course of this study. The authors also developed a mobile application for control and monitoring purposes. The control logic for high temperatures and low voltages was implemented using LEDs. An illustration of the proposed system was shown using a prototype created by the authors.

In [9], a SCADA system was implemented to manage energy in intelligent buildings. Modern buildings are equipped with several technologies that come together in this SCADA system, such as illumination, temperature, ventilation etc. In this study, a hierarchical cascade controller was implemented, which involved centralized SCADA controlling the inner loops and local PLCs controlling the outer loops. The SCADA platform was layered with a predictive controller. Based on several distributed user interfaces and energy waste minimization constraints, optimizing user preferences was achieved through the development of a predictive controller. It was demonstrated that temperature and luminosity can be controlled in large areas by using a variety of test methods. The authors also implemented a communication channel for communication between the SCADA system and the MATAB application. SCADA is suggested as a best solution for distribution network problems, such as low productivity and complexity, by the authors of [10,11]. Power electronic interfaces are critical to the development and success of emerging microgrids (MGs). The authors emphasized that the SCADA systems could improve the efficiency and productivity of active distribution networks and MGs. They used LAMBDA MG (a real microgrid used) test bed as a case study in this research and obtained real-time results from the SCADA system to demonstrate the ability of a central energy management system (CEMS) to create a proper energy balance, and thus minimize the exchange of power between LAMBDA MG and the main grid [12]. Microgrid operations could benefit from an effective SCADA system in terms of safety, reliability, and economics. Prominent use of information and communication technologies was started to enhance the operational efficiency of conventional power plants. This attracted more investment in the power sector and minimized the per unit cost of electricity [13]. In [14], the author suggested a frequency control solution by using adaptive control techniques and modeling predictive control. The findings revealed that the control algorithm converges to an optimal point, which makes the system robust.

Microgrid operations could benefit from an effective SCADA system in terms of safety, reliability, and economics. A microgrid intelligent monitoring platform was based on the SCADA systems that connect the lower central controller and upper WEB (World Wide Web) monitoring system. As a result of the SCADA system, the microgrid can be maintained in a stable and secure manner by acquiring, storing, and processing real-time data, performing load balancing and resource recovery, and establishing security at the same time [15]. According to the authors of [16], automating in a smart way can reduce costs while meeting energy demands. In residential, commercial, and industrial settings, the Internet of Things (IoT) can be used to improve energy management. They presented an open-source, low-cost, and reliable home monitoring and control system [17]. They claimed that by using analog sensors, Message Queuing Telemetry Transport (MQTT), an ESP32, and Node-RED, the proposed SCADA system was capable of allowing remote access to appliances and controlling them over local Wi-Fi.

Many researchers have investigated the architecture and different techniques of the SCADA system for commercial SCADA systems, which are expensive and far beyond the budgets of small- and medium-sized organizations. Others have designed the low-cost SCADA using an open-source WebView platform, but the security of a system is compromised because data are stored in a remote server. Therefore, a comprehensive SCADA system is designed to overcome the following issues:Provide a low-cost SCADA system that is within the range of the customer’s budget as compared to the available commercial systems.The designed SCADA system itself should consume minimum power.Data should be stored locally so that only the right person with the right authorization can access it.HMI should provide a user-friendly interface.

## 3. System Description

The designed SCADA system for a PV plant is depicted in Figure 2, which has 15 rows and 313 columns of PV modules. For demonstration purposes, only one row is being used. Two Arduino Mega 2560s (Arduino, Somerville, MA, USA), which acted as remote terminal units (RTUs), were used for taking the data from PV field sensors. In the design configuration, a low-cost Dell computer was used with Raspberry Pi software (Raspberry Pi foundation, Cambridge, UK). For this purpose, 32-bit Debian with a kernel version of 5.10 was installed on the x86 processor. The specifications of the proposed system HMI/RPI are given below:Architecture: x86;Threads per core: 02;Core per socket: 02;Sockets: 1;CPU GHz: 1.9.

After that, Emoncms, which is an open-source SCADA used for data visualization and logging, was installed on the above-stated system. The built-in scripts were used for the installation purpose on a Debian operating system. The following scripts were installed in chronological order:PHP;MQTT;REDIS;MYSQL.

It is strongly encouraged that Emoncms should be used on a dedicated machine because using it with the other programs can cause malfunctioning. Hereafter, Emoncms was configured for the local setup for which “settings.ini” has been set up, which is available in the “/var/www/emoncms/” directory.

After completing the registration and login creation, the following files given in Table 1 needed to be set up as per user requirements.

## 4. Components of the Designed System

As part of the SCADA system presented in this article, AC current, DC current, voltage, temperature, and humidity sensors collct data from the field devices. The Arduino Mega 2560 microcontroller was used, which takes data from all of the field instruments and then processes and parses the data to a Raspberry Pi-based server. After this, Emoncms takes the data from the port of the RPI and saves the logs along with the display on the computer screen (HMI). The comprehensive detail of each component is described below.

### 4.1. Sensors

Sensors are the eyes and field instruments devices (FIDs) that take the data of a physical quantity and convert it to a readable analog voltage shape for measurement purposes. Based on the physical stimulus, measurements are divided into three sections in the designed system, which are stated below:DC/PV side parameters (V_dc_, I_sc_, I_scl_);AC side measurements (V_ac_, I_ac_);Environmental factors (°C, %).

All the sensors used in the system are compatible with the industrial requirement. All selected sensors in Table 2 could easily be mounted on a DIN rail, which is the industrial standard used widely all over the world. Table 3 shows the price and power consumption of all the components. The dimensions of the DIN rail are given in Figure 3. Its top hat width was 35 mm.

#### 4.1.1. DC Current and Voltage Sensor

CR5210 was used as a DC current sensor, and CR5310 was used as a DC voltage sensor. Both provided the output DC signal in the range of 0–5 V_dc_ which is directly proportional to the input signal. It also provides the built-in isolation between the input and output sides which will save our remote terminal unit from overvoltage on the input pins. The input voltage range of CR5310 is 0–600 V_dc_ and the input current range of CR5210 is 200 A_dc_. The connection and wiring of the sensors are shown in Figure 4, and both needed a 24 V DC supply for input power because it was an industrial practice to use bit high voltage; otherwise, AC may interfere with it and cause malfunctioning.

The technical specifications of the CR5210 current sensor are given in Table 4.

#### 4.1.2. AC Current and Voltage Sensor

CR4500 was used as an AC current sensor, and CR4310 was used as an AC voltage sensor. Like the DC sensors, these also provided an output DC signal, ranging from 0–5 V_dc_ which has a direct linear relationship with the input parameter along with the 24 V power supply. CR4500 is a low-cost, fully integrated, rugged, hall effect-based linear current sensor with a current range of 0–200 A, along with isolation on both sides of the CT. Table 5 shows the voltage measuring range of CR4310 is 0–500 VAC. Figure 5 shows that the wiring of the sensor and CR4500 does not need any input supply. Only the wire-carrying current will act as a primary winding. It can be demonstrated from Table 6 that it has an accuracy of 0.75% and can measure a signal of only 50/60 Hz frequency. The detailed wiring of the sensors is given in Figure 5.

#### 4.1.3. Environmental Sensor

The temperature has a direct impact on the performance and efficiency of PV systems. Solar modules are tested under STC conditions, i.e., 25 °C, but they might be installed at a site with a much higher temperature. The higher temperature has a direct impact on the open-circuit voltage and short-circuit current, which is depicted in Figure 6.

Humidity also has a direct effect on the Direct Normal Irradiance (DNI) and Global Horizontal Irradiance (GHI) which ultimately affected the output of the PV system. Therefore, a DHT11 sensor was used for measuring both parameters. Table 7 shows that the DHT11 has a calibrated signal output and guarantees long-term stability and excellent reliability. It has a resistive type of humidity measurement component for humidity and NTC thermistor for temperature, which are connected to build an 08-bit microcontroller that provides a fast response, is good quality, and is cost saving. The configuration of the DHT11 with the microcontroller unit is shown in Figure 7.

### 4.2. Remote Terminal Unit (Arduino Mega 2560)

The Arduino Mega is an Atmega2560-based microcontroller and is widely used due to its low-cost and low-power consumption. It has 54 digital input/output pins, of which 15 can be utilized as a PWM generator. In addition, it has 4 UARTS, 16 input analog pins, a 16 MHz crystal oscillator, an in-circuit serial programming (ICSP) header, and a reset button. Its input voltage limit is 7–20 V which makes it feasible to power up using a USB cable, lithium cell, or any battery. It has a flash memory of 250 KB, its static RAM has a size of 8 KB, and its EEPROM is 4 KB. The detailed pin configuration and layout of the RTU are shown in Figure 8.

The Arduino Mega is programmed using the Arduino compiler Integrated Development Environment (IDE). The program is written in Arduino IDE using its language and libraries, which are almost similar to the C/C++ language. Arduino is connected with to serial port of a PC using a type A/B cable for programming and communication. Firstly, the Arduino program, called “sketches”, are written in IDE to measure the field instruments parameters. Then, the values of these parameters are observed using the specific baud rate, which is 38,400 bits per second in our case.

### 4.3. Master Terminal Unit (RPI)

Raspberry is a small single-board computer that has been modified through several versions and features. It has many applications ranging from small toys to industrial control. It requires an operating system (OS), usually Raspbian, to carry out the operations. The main flexibility of the MTU is that any computer can also be transformed into the RPI as per need. In the designed case, an old core i3 computer was used as the RPI and Debian was used as the OS for this PC. It had a 15.6′’ screen and a processor with the following specifications:A 1.90 GHz quad-core x86 Intel CPU;500 GB ROM;4 GB RAM;One hundred Base Ethernet;Three USB ports (2 USB 3.2 and 1 USB 2.0);Full HDMI port;Micro SD card slot;VideoCore IV 3D graphics core.

### 4.4. Emoncms

As it has already been discussed, Emoncms is an open-source platform and provides versatile packages as per user demand. The main objective is to display the data on an HMI to visualize and observe the behavior of a system. It also offers an easy package where an account can be created on the Emoncms website and data is stored on a remote server, which limits the data-storing capabilities and also exposing that data to hackers, which compromises the security and reliability of the system. Therefore, emonPi was used to store the data locally and acted as a local server whose specifications are shown in the Figure 9.

Emoncms is itself a WebView platform but it needs many other modules and data in a specific format called JSON to display on the HMI. Below, Figure 10 explains Emoncms’s interconnectivity of different modules, in which the main module is Emonhub. Emonhub takes data from an RTU in its raw form and refines it using the MQTT module. Mosquito can also be used instead of MQTT for data formatting. Emoncms WebView needs PHP, MYSQL, and REDIS for data visualization and logging. If any one of them is not working, there will be an error. The source code for Emoncms configuration can be found in [18].

## 5. Implementation Methodology

In the implementation of the designed system, voltage sensors were connected in parallel and current sensors were connected in a series. The DC transducers were connected on the analog pins A0, A2, and A4 of RTU1 (Arduino Mega 1), and AC transducers were connected on the analog pins A0 and A4 of RTU2 (Arduino Mega 2). In addition, DHT11 was also connected to the Tx and Rx pins of RTU2. Below, Algorithm 1 represents the pseudocode of Arduino, which was compiled and built in the Arduino chip. The source code can be found in Appendix A, Algorithm A1.
**Algorithm 1: Arduino Sensor Data Reading Algorithm**
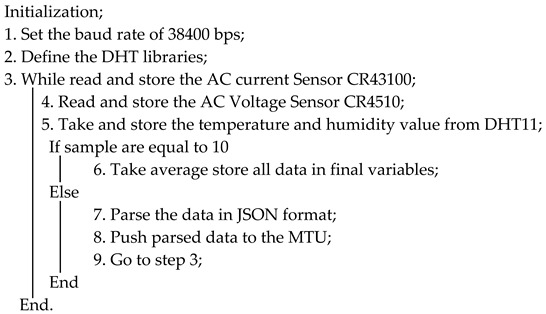


Algorithm 2, along with the below Figure 11 flowchart, represents the complete SCADA system from the RTU to the HMI. The source codefor Algorithm 2 is presented in Appendix A, Algorithm A2.
**Algorithm 2: Data Logging Algorithm**
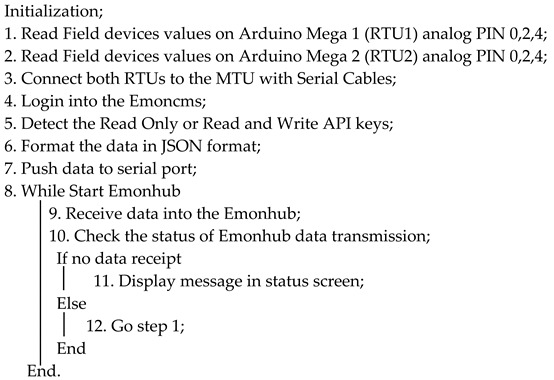


## 6. Prototype Design and Results

The proposed SCADA system setup is shown in Figure 12 using the above-stated hardware and operating principles. Figure 12 shows that the current and voltage transducers are connected to the corresponding Arduino, and both are connected with the MTU, which is a Dell laptop. Different power supplies were used to generate the desired voltage. A 24 V supply was used for proving the V_cc_ and ground to all sensors. To provide the DC voltage, a variable supply was used with an output voltage that varied between 0–60 V_dc_. A 12 V DC incandescent bulb and rheostat were used as a DC load to generate the load current. On the AC side, an autotransformer was used which provided isolation and protection from the grid along with the variable voltage. A 100 W incandescent bulb was used on the AC side as a load. The physical fluke voltmeters and current meters were installed on point of interest to cross-check the measurements of field parameters. After connecting all of the field devices to the RTU, both Arduinos were connected to the RPI, using a USB 2.0 Cable Type A/B, which provided the communication.

The system was turned on for 6 h and data logging was performed. During the testing of the system, different digital multimeters were connected to points of interest to observe the value of the parameters. The values calculated with the designed setup were very close to the actual multimeter value and had an accuracy error of less than 5% over the entire measurement range. The designed system is capable of storing and logging PV system data for months or years.

Various dashboards were created to observe the trend of different values and also to create a user-friendly environment for observations. Figure 13 shows the dashboard for observing the instantaneous values of PV arrays, the inverter output, and environmental factors.

In Figure 14, the data was logged and a graph was plotted from that data. The below figure shows the inverter output current, phase voltage, temperature, humidity, and the display of all parameters on one graph for a period of 6 h. The recording period can be adjusted using the tab just above the graph

Figure 15 demonstrates the PV array parameters. It depicts the behavior of the total PV current, PV voltage, and Array 1 current. This data was logged for 6 h for basic testing, but the system is capable of logging data as long as needed without any modification or addition.

The emonPi provided both local and remote data logging using the Emoncms web application display, which maintains the data with the privacy of your home. A total of 10 GB storage was enough for a node with six feeds to store the data. The designed system also provides the least possible wear out of memory, which ultimately increases the lifetime of the hard disk. More sensors could be added, such as logging the PV current for each PV string.

## 7. Discussion

The following are a few key features of the designed system, which was verified through successful testing:Emoncms-based SCADA system: Using the Emoncms platform, the system offers all of the features common to most recent SCADA systems, such as data acquisition, data transmission, remote terminal units (RTU), and master terminal units (MTUs).Data monitoring: Monitoring data remotely is possible through the WebView feature of the system.Low-cost and open-source components: Low-cost and open-source components contribute to the overall low cost of the proposed SCADA system with all the features of modern, expensive commercial systems.Data storage: It is also possible to view the data trends for any time period, from one hour to one year.User friendly dashboard: Data monitoring is made easier with the dashboard’s user-friendly interface.No license fee: The designed system has no license fee or yearly fee to cover software.Private and secure system: Due to local data storage, the design system is not sending data to any remote server. It is 100% private and secure.

## 8. Conclusions

Most of the industries in the world are located in remote areas, due to the large land requirements and each country’s industrial legislation. Therefore, it is compulsory to have a local source of data logging to calculate the monthly and annual costs and return on investment, along with the cost effectiveness and reliability. Indeed, the SCADA was modified through many tiers from the distributed control system to the IoTs, but mostly, it remained proprietary and costly. As in power system applications, batteries, inverters, PV system, etc. are purchased from different manufacturers, which give interfacing and compatibility issues during the monitoring and control. Thus, an open-source SCADA is an inevitable solution that permits “mix and match” components from different manufacturers. In addition, proprietary SCADA is very costly, and only major companies with big revenues can afford this. In such a situation, an open-source Emoncms SCADA system is the best, most reliable, cost-effective solution, and a 100% private and protected system.

In the designed system, the low-cost, open-source SCADA system was designed with IoTs architecture, which is the latest one. The initial capital cost of the whole SCADA system was CAD $761.72, and the running cost of the system was zero because it is open source and storing the data in the local server. The designed system’s overall cost was less than any system studied in the above literature or mentioned on the supplier’s website. For example, the system “NOVUS 8845000080 SuperView SCADA Software (ITM Instruments Inc., Sainte-Anne-de-Bellevue, QC, Canada)” costs CAD $911, but including the hardware, the cost will be much higher.

The hardware design of the system was also performed and had four major modules: field instrumentation devices (current, voltage, and temperature sensors), remote terminal units (Arduino Mega microcontroller), master terminal units (RPI and Emoncms), and the SCADA communication channel (local communication through serial port). To test the system, it was set up in the lab and different voltage and current sources were used to observe the behavior of the system. The system voltage and current were changed and a corresponding change in the output was also observed. Additionally, conventional digital multimeters were used to cross-check the parameters of the system. The system also stored the data locally and created dat and meta files in the “/var/opt/emoncms/phpfina” directory. From testing, it was deduced that the designed system could work in a robust environment and log the data in a real-time environment with great accuracy and precision. Furthermore, the power consumption of the whole designed SCADA system was less than 35 W, including the HMI display which consumed significant power, i.e., a 30 W consumption. The design system’s total cost was only CAD $761.72, which is the least cost of an open-source SCADA system with industrial sensors, while commercial SCADA has a proprietary software license fee; an annual fee running into thousands of dollars per year.

## Figures and Tables

**Figure 1 sensors-22-06733-f001:**
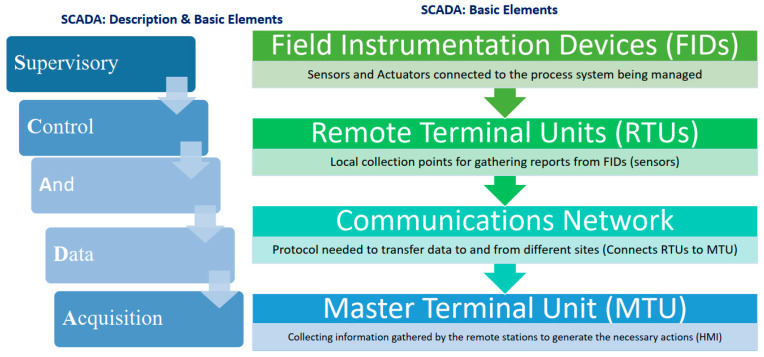
System components.

**Figure 2 sensors-22-06733-f002:**
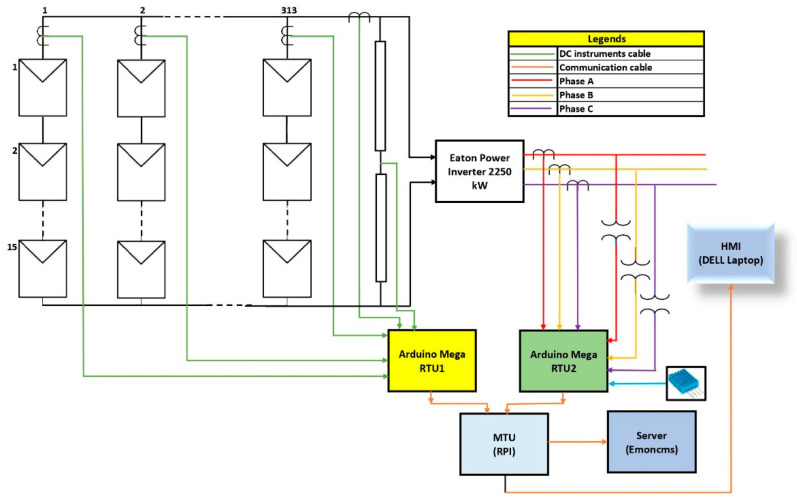
Schematic of PV monitoring system.

**Figure 3 sensors-22-06733-f003:**
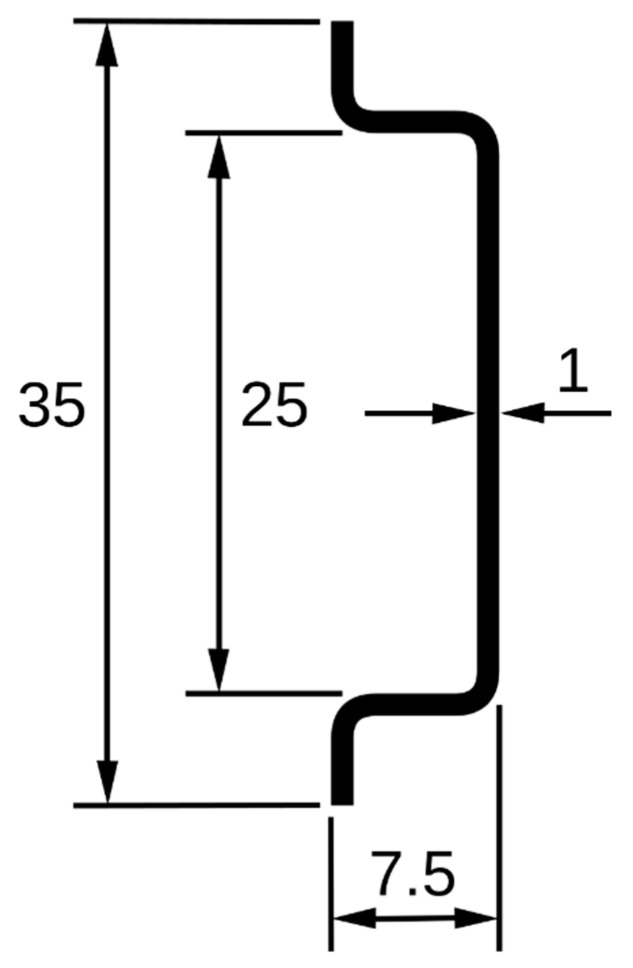
Dimensions of the DIN rail.

**Figure 4 sensors-22-06733-f004:**
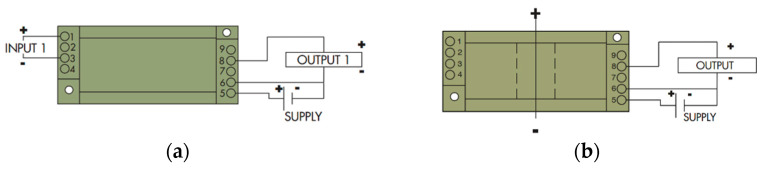
Wiring of the DC transducer: (**a**) Voltage sensor CR5310; (**b**) Current sensor CR5210.

**Figure 5 sensors-22-06733-f005:**
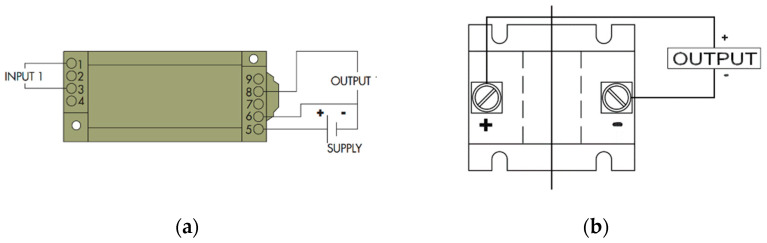
Wiring connection of the AC transducers: (**a**) AC voltage sensor CR4310; (**b**) AC current sensor 4500.

**Figure 6 sensors-22-06733-f006:**
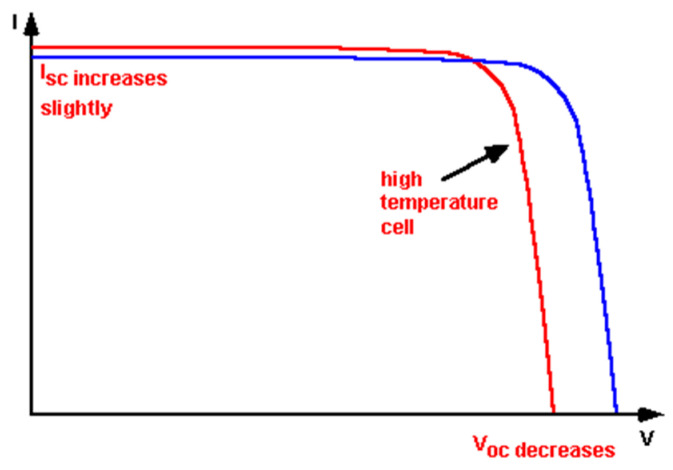
Effect of temperature on PV module characteristics.

**Figure 7 sensors-22-06733-f007:**
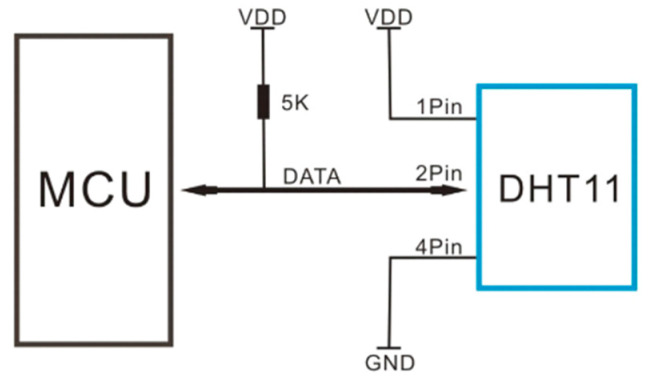
DHT11 interfacing with Arduino.

**Figure 8 sensors-22-06733-f008:**
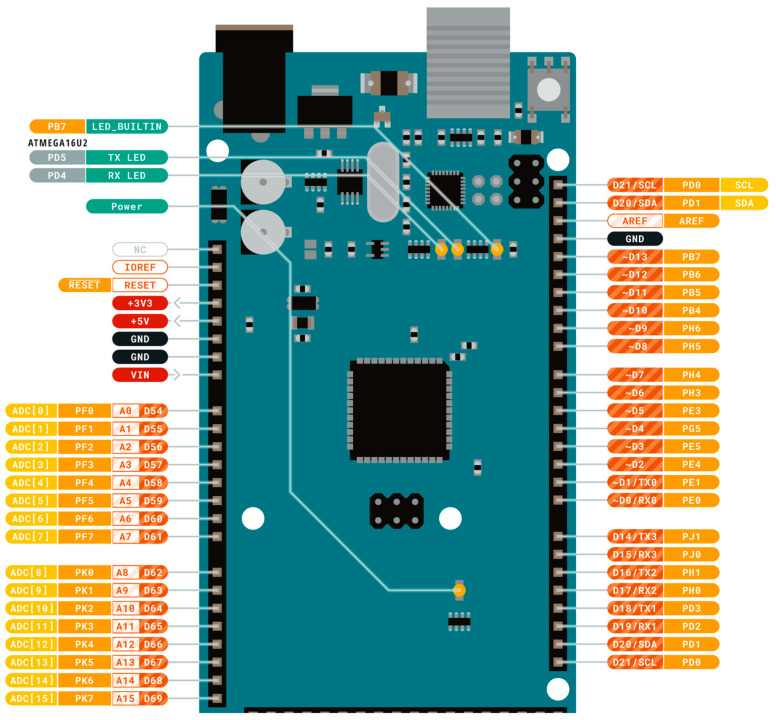
Arduino Mega 2586 pin layout.

**Figure 9 sensors-22-06733-f009:**
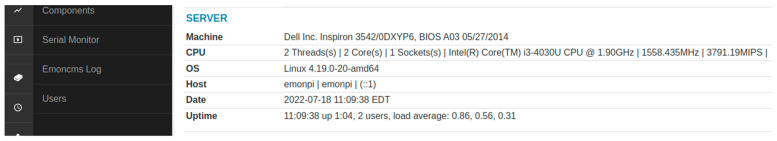
Dell laptop as local server.

**Figure 10 sensors-22-06733-f010:**
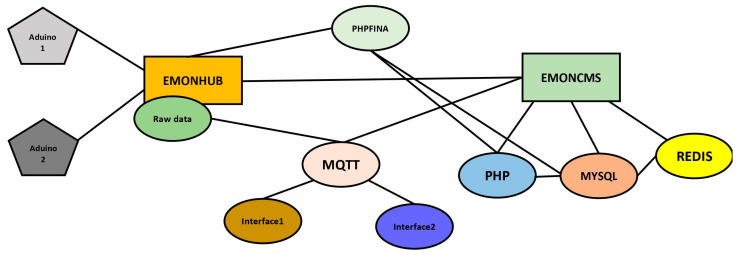
Emoncms connectivity modules.

**Figure 11 sensors-22-06733-f011:**
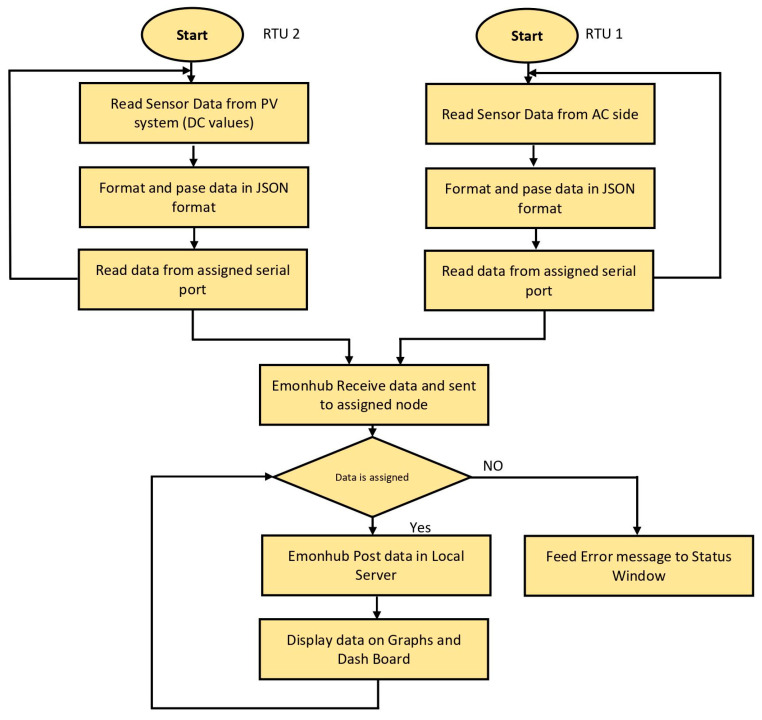
Flowchart of complete data processing.

**Figure 12 sensors-22-06733-f012:**
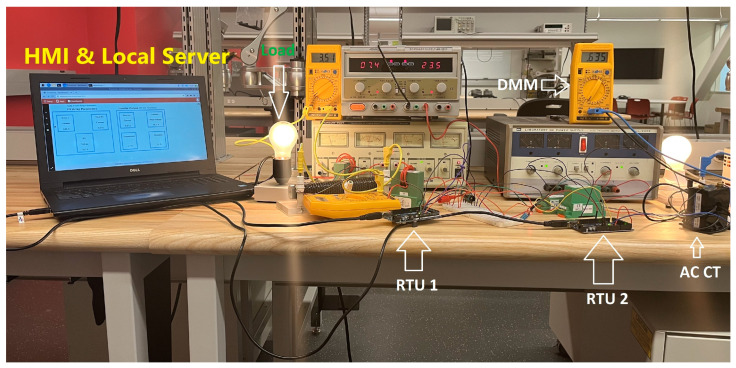
Hardware setup.

**Figure 13 sensors-22-06733-f013:**
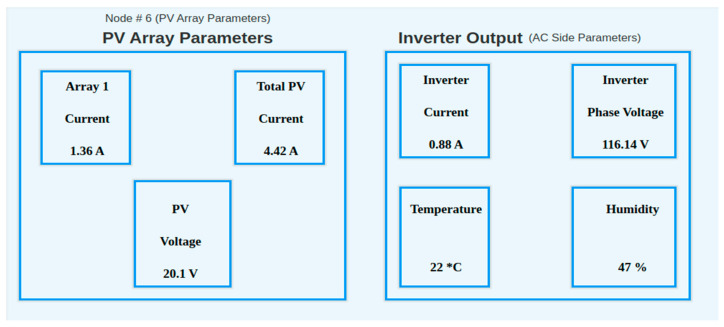
Instantaneous values of field instruments.

**Figure 14 sensors-22-06733-f014:**
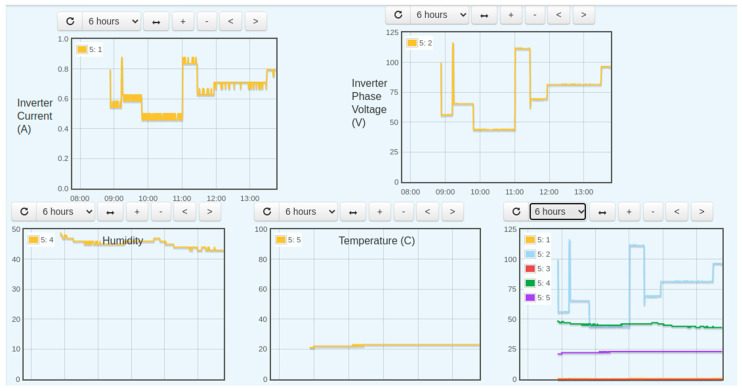
AC side parameters trend.

**Figure 15 sensors-22-06733-f015:**
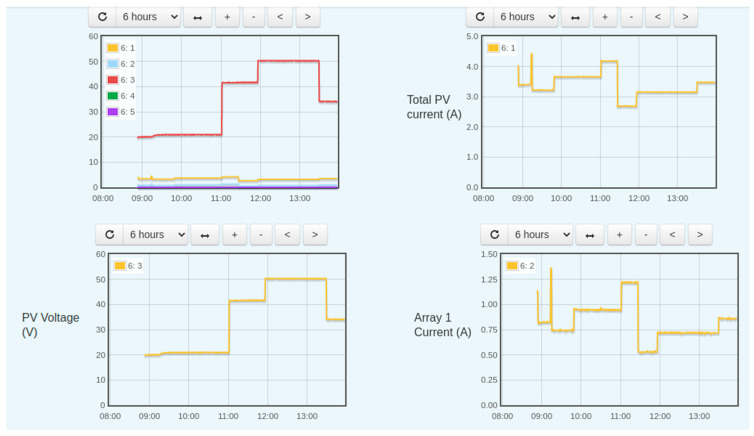
DC side parameters behavior.

**Table 1 sensors-22-06733-t001:** Setup file paths.

Function	Directory
Data	/var/opt/emoncms
Location of main code	/var/www/emoncms
Install location for modules symlinked to www	/opt/emoncms
Log file Directory	/var/log/emoncms
Installation location of Emonscripts codes	/opt/openenergymonitor

**Table 2 sensors-22-06733-t002:** Sensor models.

Sr.#	Manufacturer	Model	Function
1	CR Magnetics	CR5210	DC Current Transducer
2	CR Magnetics	CR5310	DC Voltage Transducer
3	CR Magnetics	CR4510	AC Voltage Transducer
4	CR Magnetics	CR4310	AC Current Transducer
5	Aosong Electronics	DHT11	Temperature and Humidity

**Table 3 sensors-22-06733-t003:** Sensor power consumption and price.

Sr.#	Manufacturer	Model	Price ($)	Power Consumption (W)
1	CR Magnetics	CR5210	116.26	0.84
2	CR Magnetics	CR5310	190.07	0.84
3	CR Magnetics	CR4510	210.40	0.36
4	CR Magnetics	CR4310	99.99	0
5	Aosong Electronics	DHT11	5.00	0.0015
6	Arduino	Mega 2560	70.00	0.27
7	Raspberry Pi	Dell	70.00 $	30 W

**Table 4 sensors-22-06733-t004:** CR5210 Specifications.

Sr.#	Specifications	Value	Units
1	Basic Accuracy	1	%
2	Linearity	10–100 FS	%
3	Thermal Drift	500	PPM/°C
4	Operating Temperature	0–50	°C
5	Max Response time	250	ms

**Table 5 sensors-22-06733-t005:** CR4310 Specifications.

Sr.#	Specifications	Value	Units
1	Basic Accuracy	0.5	%
2	Calibration	True RMS sensing	--
3	Frequency Range	20–5000	Hz
4	Operating Temperature	0–60	°C
5	Supply Voltage	24 ± 10%	Vdc

**Table 6 sensors-22-06733-t006:** CR4500 Specifications.

Sr.#	Specifications	Value	Units
1	Basic Accuracy	0.75	%
2	Calibration Signal out	0–5	Vdc
3	Frequency Range	50/60	Hz
4	Operating Temperature	−30 to 60	°C
5	Insulation class	600	V

**Table 7 sensors-22-06733-t007:** DHT11 Specifications.

Sr.#	Specifications	Temperature	Humidity
1	Resolution	1 °C	1% RH
2	Accuracy	±4 °C	±4% RH
3	Range	0–50 °C	20–90% RH
4	Response Time	6 s	10 s

## Data Availability

We have provided all the data related to this study in Appendix A.

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
