# Peer review of "Low-Cost, Open-Source, Emoncms-Based SCADA System for a Large Grid-Connected PV System"

_sensors, 2022, doi:10.3390/s22186733_

Round 1
Reviewer 1 Report
Thank you for offering me the opportunity to review this manuscript. The paper is presenting and argue the advantage of the proposed system.
I suggest to add a paragraph inn the conclusions section about costs (initial), running costs, in comparison with the existing systems.
Author Response
Dear Reviewer,
We appreciate your permission to resubmit our manuscript and thank you for giving us the opportunity to respond to the reviewers' comments. Reviewer 1 advised us to address,
- I suggest adding a paragraph inn the conclusions section about costs (initial), running costs, in comparison with the existing systems.
As advised Some addition has been made in the conclusion and is presented below. Same has been included in the original manuscript.
The initial capital cost of the whole SCADA system is CAD $ 761.72, and the running cost of system is zero because it is open source and storing the data in the local server. The designed system overall cost is less than the any system studied in the above literature or mentioned on supplier’s website. For example, system “NOVUS 8845000080 SuperView SCADA Software” costs CAD $ 911 and including hardware the cost will be much higher.
The design system's total cost is only CAD $ 761.72 which is the least cost of an open-source SCADA system with industrial sensors while commercial SCADA has proprietary software license fee, annual fee running into thousands of dollars per year.

Reviewer 2 Report
In this paper, the authors describe a low-cost Supervisory Control and Data Acquisition system for a PV plant. It is interesting that the authors carried out experiments in order to validate the efficiency of the proposed system. The paper is clearly written and well-organized. Some points should be included within the manuscript in order to improve the publication.
· The abstract should be also rewritten in a more compact and successive way.
· The authors should describe in more detail the experimental setup and the experimental procedure in section 6. The authors should specify the characteristics of the components of the experimental setup.
· The authors should explain the contribution of their study in comparison with the corresponding experiments and analysis of other researchers.
· The technical contribution of the current work is not clear. The authors do not clearly clarify their contributions in the abstract and conclusions section.
· Please introduce discussions with other articles in your conclusions. Provide two sample articles related to smart grids:
o L. Priya, V. Gomathi, "Demand Response Management Algorithm for Distributed Multi-Utility Environment in Smart Grid," WSEAS Transactions on Power Systems, vol. 15, pp. 230-239, 2020.
o Adel Elgammal, Tagore Ramlal, "Optimal Model Predictive Frequency Control Management of Grid Integration PV/Wind/FC/Storage Battery Based Smart Grid Using Multi Objective Particle Swarm Optimization MOPSO," WSEAS Transactions on Electronics, vol. 12, pp. 46-54, 2021.
Author Response
Re: Response to reviewer 2
Dear Editor,
We appreciate your permission to resubmit our manuscript and thank you for giving us the opportunity to respond to the reviewers' comments. The reviewer 2 advised to address:
- The abstract should be also rewritten in a more compact and successive way.
As advised the abstract is rewritten and refined, in more compact form.
- The authors should describe in more detail the experimental setup and the experimental procedure in section 6. The authors should specify the characteristics of the components of the experimental setup.
As advised some more details are added in section 6. Characteristics of components are presented in section 4 of this article.
- The authors should explain the contribution of their study in comparison with the corresponding experiments and analysis of other researchers.
As advised, contributions are added in the conclusion section and as a part of the discussion section to make it more understandable to the researchers.
- The technical contribution of the current work is not clear. The authors do not clearly clarify their contributions in the abstract and conclusions section.
Contributions are added in the conclusion section and as a part of the discussion section to make it more understandable to the researchers.
- Please introduce discussions with other articles in your conclusions. Provide two sample articles related to smart grids:
Some sample articles are added relating to smart grids and discussion section of this articles has been enhanced with some more information.

Reviewer 3 Report
This research is establishing a low-cost Supervisory Control and Data Acquisition (SCADA) system for a PV plant. Data logging is performed by an inverter output that provides data to RTU2 and the Raspberry Pi in JSON format. The manuscript is written well, but some modifications and arrangements are required:
1- The introduction is briefly described and not sufficient for the readers. Add more references related to the SCADA system of different methodologies in the introduction section.
2- Some abbreviations are described multiple times, and some are not. Add nomenclature at the end of this manuscript that fulfills this query.
3- It is said that low-cost SCADA is in the title, but no comparison is presented in the manuscript. Justify why. It is recommended to show the comparison with other studies.
4- Presented figures are not visible clearly and have low quality; the quality of figures needs to be improved, enhanced all the figures.
5- It is recommended to provide the source code for readers that assist them in building their own.
6- Provide what kind of data is saved in the .dat extension in the form of a Table under the appendix.
Author Response
Re: Response to reviewer 3
We Thank You for your permission to resubmit our manuscript and thank you for giving us the opportunity to respond to the reviewers' comments. The reviewer 3 advised to address
- The introduction is briefly described and not sufficient for the readers. Add more references related to the SCADA system of different methodologies in the introduction section.
The articles using different methodologies are presented as a part of the literature review such as reference 7, 8 and 9.
2- Some abbreviations are described multiple times, and some are not. Add nomenclature at the end of this manuscript that fulfills this query.
List of abbreviations used as a part of this research has been added as advised.
3- It is said that low-cost SCADA is in the title, but no comparison is presented in the manuscript. Justify why. It is recommended to show the comparison with other studies.
The research work carried throughout this article is compared with the commercially available software (that does not include components cost) following the reviewers comments. Details are added in the conclusion.
4- Presented figures are not visible clearly and have low quality; the quality of figures needs to be improved, enhanced all the figures.
All figures have been replaced by the improved quality figures.
5- It is recommended to provide the source code for readers that assist them in building their own.
The data availability statement will be added in the final version of the paper for future participants of such kind of research.
6- Provide what kind of data is saved in the .dat extension in the form of a Table under the appendix.
The data in single column format. The details are on line 432-434 in the conclusion section.

Round 2
Reviewer 3 Report
Question 5 still not clearly presented by authors. Do not confuse the reviewer by just saying that it is fulfilled. Again asking for source code in raw format as well as a link.
Author Response
Re: Response to reviewer 3- Round-2
We appreciate your guidance and are thankful for giving us a chance to update the manuscript with this valuable information. Reviewer 3 advised us to;
- Question 5 still not clearly presented by authors. Do not confuse the reviewer by just saying that it is fulfilled. Again asking for source code in raw format as well as a link.
We have added the link for the complete source code of Emoncms configuration. The reader can see the raw code and details to setup Emoncms. Reference [18].
And the source code of both the algorithms presented as a part of this study are presented in Appendices.
We hope this works and welcome any further suggestions for the improvement of this research work.
Thank You
